# Disparity of cycad leaves dispels the living fossil metaphor
Mario Coiro [1,2] ✉ & Leyla Jean Seyfullah [1]

The living fossil metaphor is tightly linked with the cycads. This group of gymnosperms is supposed to be characterised by long-term morphological stasis, particularly after their peak of diversity and disparity in the Jurassic. However, no formal test of this hypothesis exists. Here, we use a recent phylogenetic framework and an improved character matrix to reconstruct the Disparity Through Time for cycad leaves using a Principal Coordinate Analysis and employing Pre-Ordination Ancestral State Reconstruction to test the impact of sampling on the results. Our analysis shows that the cycad leaf morsphospace expanded up to the present, with numerous shifts in its general positioning, independently of sampling biases. Moreover, they also show that Zamiaceae expanded rapidly in the Early Cretaceous and continued to expand up to the present, while now-extinct clades experienced a slow contraction from their peak in the Triassic. We also show that rates of evolution were constantly high up to the Early Cretaceous, and then experienced a slight decrease in the Paleogene, followed by a Neogene acceleration. These results show a much more dynamic history for cycads, and suggest that the 'living fossil' metaphor is actually a hindrance to our understanding of their macroevolution.

The idea of living fossils is as old as evolutionary thought itself, being introduced by Darwin[1] to refer to species or groups that have experienced minimal change across time, and thus closely resembled their fossil ancestors. Since its inception though, various authors have employed different criteria to classify organisms as living fossils. While most scholars emphasize morphological stasis, others stress different criteria such as lack of species diversity, persistence of a lineage through geological time, phylogenetic uniqueness, geographically restricted distribution, and others, leading to a confusing and vague definition[2–4]. Surprisingly though, this metaphor still holds substantial power in swaying research programmes and generating hypotheses[5].

Cycads are a charismatic group both for scientists and the general public. They are a group of gymnosperm plants characterised by a palm-like habit and large compound leaves, as well as having plants of separate sexes (dioecy). The order Cycadales includes around 375 species in two families[6], the Cycadaceae (including the genus *Cycas* L. and ~ 119 species) and the Zamiaceae (including the other 9 genera and the majority of the species diversity). Nowadays, they are distributed in tropical and subtropical climates, with major centres of diversity in Mexico and Central America, South Africa, and Australia[7]. Among the extant plants, the diversity and disparity of cycads is dwarfed by other groups such as the angiosperms and the polypodiaceous ferns, but it is similar to that of other groups of comparable age, such as Araucariales (around 200 species)[8] or Gleicheniales (also around 200 species)[9].

Even so, cycads are often considered together with Ginkgoales (including the only extant species *Ginkgo biloba* L.) as examples of plant living fossils.

Indeed, cycads' reputation as "dinosaur plants", apparently unchanged since their origin in the Palaeozoic and dominance during the Mesozoic, still dominates the discourse surrounding this group[10–12]. Although the results of molecular and total-evidence phylogenies indicate that in terms of species diversity, the cycads are a rather young group, diversifying in the late Cenozoic[12,13], they are still considered to be morphologically similar, if not identical, to their fossil relatives[14]. Their peak of diversity and morphological disparity is supposed to be in the Jurassic (201.4–145 Ma), followed by a decline leading to a depauperate modern flora[15].

Recent discoveries and analysis have challenged this view, indicating a much complex pattern of morphological evolution than expected from the living fossil metaphor. Cycad fossils with unexpected morphology, such as the diminutive male cone from the Early Cretaceous of California, suggest the presence of much-hidden diversity of Zamiaceae during this period, possibly indicating a radiation[16]. This pattern is further supported by phylogenetic analyses including fossil cycad leaves[17]. This analysis found a geographical expansion of the Zamiaceae across the Jurassic and the Cretaceous, corresponding with the appearance of fossils that can be confidently assigned to this family. Taken together, these results suggest that the dynamics of cycad diversification and disparification might be complex, and that the living fossil metaphor might be a detrimental constraint on cycad research.

[1]Department of Palaeontology, University of Vienna, Vienna, Austria. [2]Ronin Institute for Independent Scholarship, Montclair, NJ, USA.
✉e-mail: mar.coiro@gmail.com

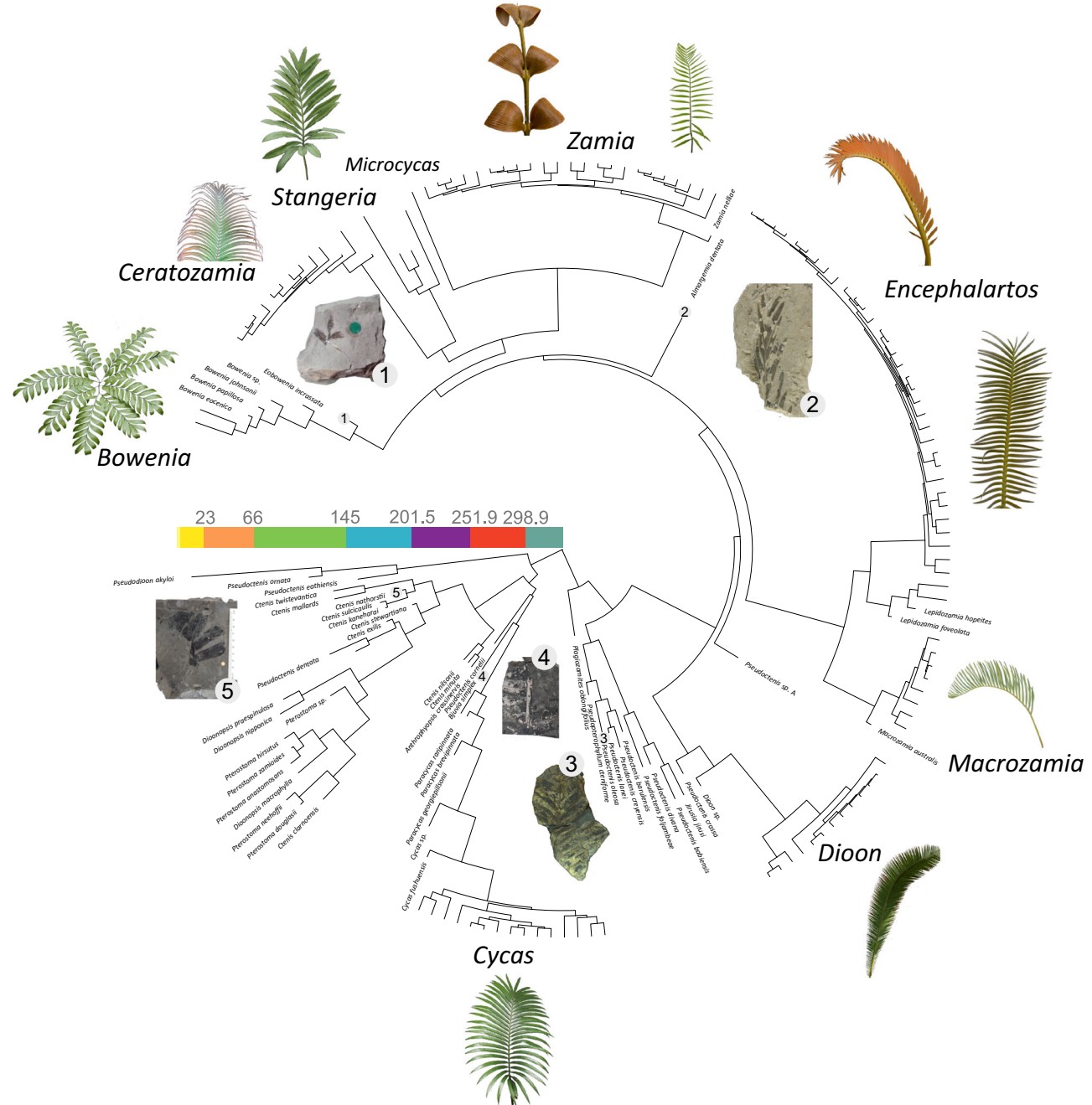

**Fig. 1 | Trimmed time-calibrated consensus tree from Coiro et al.[17] used in this study.** Leaves of *Bowenia spectabilis* Hook. ex Hook.f., *Ceratozamia chimala-pensis* Pérez-Farr. & Vovides, *Stangeria eriopus* (Kunze) Baill., *Zamia imperialis* A.S.Taylor, J.L.Haynes & Holzman, *Zamia* sp., *Encephalartos lehmannii* Lehm., *Encephalartos inopinus* R.A.Dyer, *Macrozamia secunda* C.Moore, *Dioon edule* Lindl., and *Cycas thouarsii* R.Br. are shown as examples of extant cycad leaf diversity. Fossils of *Eobowenia incrassata* (S.Archang.) M.Coiro & C.Pott (1), *Almargemia dentata* Florin (2), *Pseudoctenis oleosa* Harris (3), *Bjuvia simplex* Florin (4), and *Ctenis nathorstii* Möller (5) are shown as examples of cycad fossil leaf diversity. Images are not to scale. Extant cycad images courtesy of Michael Calonje, except *Dioon edule* who has been taken by the authors. Image of *Almargemia dentata* from Coiro & Pott[21], used under a CC BY 4.0 license.

Here, we use a recent phylogenetic hypothesis for cycad leaf fossils (Fig. 1) to conduct the first formal analysis of the macroevolutionary dynamics of leaf disparity in the Cycadales by reconstructing leaf disparity through time. We show that cycad disparity has increased through time, mostly due to the origin and expansion of the Zamiaceae after the Jurassic Period, and that the rates of evolution have not declined up to the present but instead show a recent increase. This presents a much more dynamic history for this group of plants, and suggests that the living fossil metaphor should be abandoned in favour of more productive ones.

## Results

Principal Coordinate Analysis of the morphological matrix resulted in a matrix of 337 rows, corresponding to 169 tips, 168 internal nodes and 335 columns (PCoA axes). A Scree plot (Supplementary Fig. 1) shows that the first 10 axes account for around 10% of the variation, compatible with a similar analysis of other matrices[18].

The only taxa (OT) morphospace and the Pre-Ordination Ancestral State Reconstruction (POASR) morphospace show almost identical distributions (Fig. 2). PC1 separates Zamiaceae from the rest of the cycad leaves, and PC3 separates the leaves of Cycadaceae from the rest, while PC2

## Only Taxa morphospace

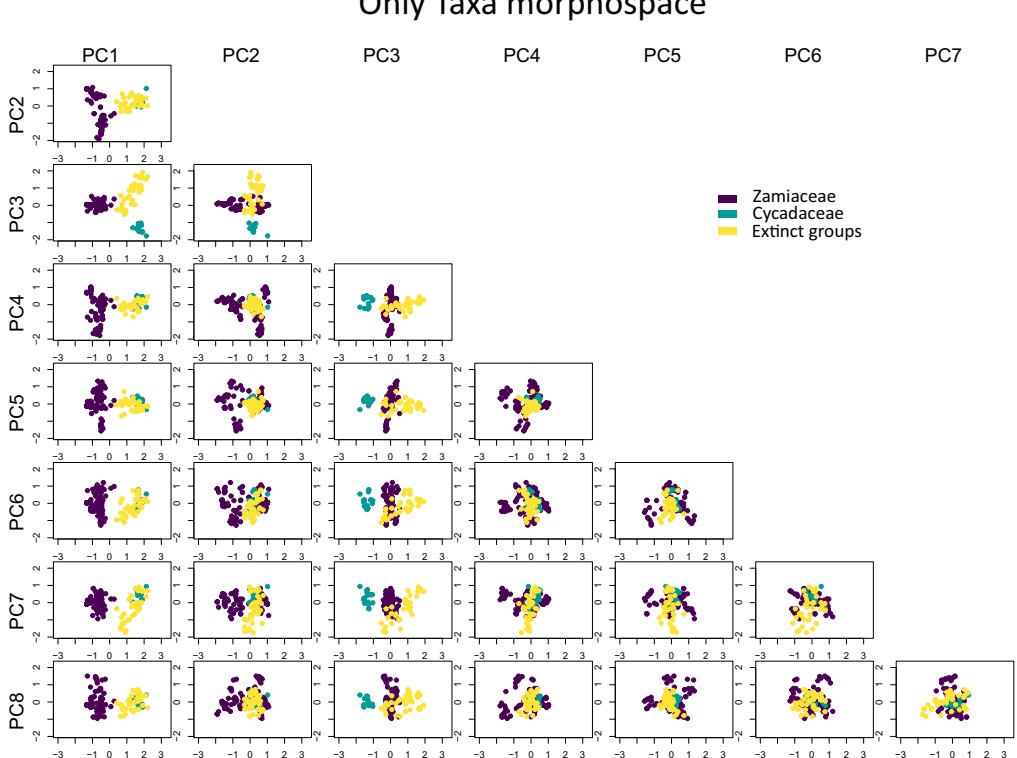

## Pre-Ordination Ancestral State Reconstruction morphospace

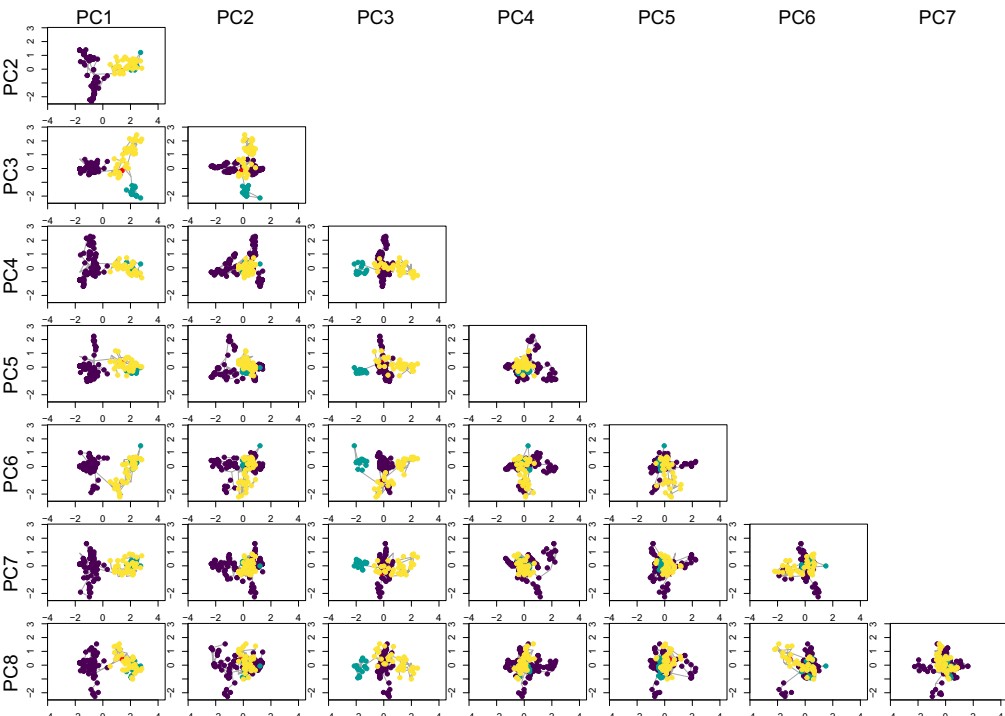

**Fig. 2 | Scatter plots for the first eight Principal Coordinate Analysis axes (PC1-PC8) for the Only Taxa morphospace (top) and the Pre-Ordination Ancestral State Reconstruction morphospace (bottom).** Tips and nodes for Zamiaceae are coloured in purple, Cycadaceae in green, and extinct cycads in yellow. The figure shows that the first PcoA axis (PC1) separates Zamiaceae from the rest of the cycads, and that in general the three groups of cycads occupy different parts of the morphospace. On the bottom graphs, the root node is coloured in red, and the phylogenetic relationships are indicated with grey lines.

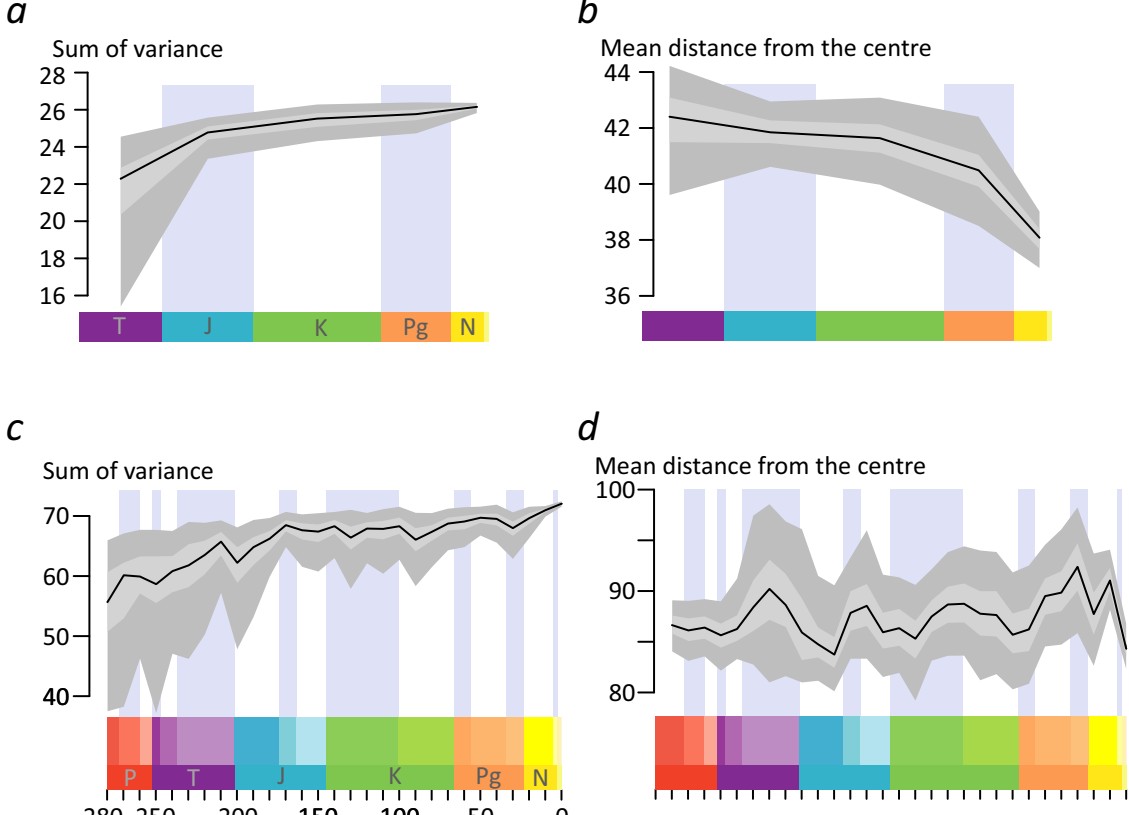

**Fig. 3 | Disparity-Through-Time (DTT) plots of cycad leaves, showing a more dynamic pattern than expected from stasis and/or post-Jurassic decline. a** DTT plot of the sum of the variance of the 'OT' analysis, i.e. including only the tips of the cycad tree, showing an increase of the morphospace of cycad leaves up to the present. **b** DTT plot of the mean distance from the centre of the 'OT' analysis, showing the placement of the morphospace moving through time up to the present. **c** DTT plot of the sum of variance of the Pre-Ordination Ancestral Reconstruction analysis. **d** DTT plot of the mean distance from the centre of the pre-ordination ancestral reconstruction analysis. Both **c** and **d** show a similar but more complex pattern than **a** and **b**. Black line indicates the bootstrapped median, light grey indicates the 25%-75% percentiles, dark grey indicates the 2.5–97.5% percentiles.

has a less clear phylogenetic signal. Both sums of variance and the mean distance from the centre do not show strong sensitivity to rarefaction neither in the OT analysis (Supplementary Fig. 2) nor the POASR analysis (Supplementary Fig. 3), with the exception of some of the bins with the smaller sample size.

## Disparity through time

In the OT analysis, the size of cycad leaf morphospace, expressed as the sum of variance, shows an initial increase between the Triassic and the Jurassic (Fig. 3a). Although the increase slows down, it continues up to the present, with a small plateau between the Cretaceous and the Paleogene. The mean distance from the centre, indicating the position of leaf morphospace, shows smaller shifts between the Triassic and the Cretaceous, followed by larger shifts towards the Paleogene and the Neogene-Quaternary (Fig. 3b).

In the analysis using data reconstructed for the nodes, the sum of variance shows an initial expansion during the Permian and the Triassic, with a peak around 210 Ma followed by a drop (Fig. 3c). A subsequent increase up to a peak in the Mid Jurassic (170 Ma) is followed by oscillations leading to overall stasis. After a dip during the Late Cretaceous, a slower increase leads to a further peak in the Eocene, followed by a drop in the Oligocene and a recovery up to the highest levels of disparity in the present.

The position of the morphospace is also quite dynamic, with major shifts in distance from the centre in the transition between the Early and Late Triassic, Late Triassic and Early Jurassic, Early Jurassic and Mid Jurassic, Late Jurassic and Early Cretaceous, Late Cretaceous and the Oligocene, as

well as Oligocene to early Miocene, early to late Miocene, and late Miocene to the present (Fig. 3d). The difference between the mean distance to the centre in the present and at the origin shows a shift in the position of the extant morphospace compared to the original morphospace, similar to the tip only analysis. The results from the POASR analyses are also robust to topological uncertainty (Supplementary Fig. 4).

## Clade-specific disparity through time

Portioning the disparity according to taxonomic grouping shows that while the Mid Jurassic peak is driven by extinct taxa, the subsequent growth of leaf morphological space is mostly driven by the expansion of Zamiaceae (Fig. 4a, Supplementary Fig. 5). This group shows a first period of expansion up to the Early Cretaceous, with a peak at 120 Ma, followed by two other major periods of expansion between the Late Cretaceous and the Eocene and between the Miocene and the present. The space occupied by extinct taxa shows a progressive shrinkage from a Mid Jurassic peak (Fig. 4b, Supplementary Fig. 6). Smaller peaks are found in between the Early and Late Cretaceous and in the late Eocene. The space reaches its smallest extent before the complete extinction of the *Ctenis* clade.

## Rate analysis

Point rate estimates for our selected time bins show the highest rates in the Carboniferous, with a slowdown leading to a minimum in the Jurassic and then a rise in the Early Cretaceous (Fig. 5). Rates decrease in the Late Cretaceous and Paleogene, only to jump up during the Neogene-Present. However, AICc selection favours a much simpler model, with a single rate of 0.07 character changes per Ma from the Carboniferous to the Early

Cretaceous, a lower rate of 0.05 in the Late Cretaceous and Paleogene, and the highest rate of 0.09 during the Neogene and Present.

The analysis of rates in different clades corresponding to the acquisition of nitrogen fixation favored a model with a single rate across the tree (Fig. 6). However, a model with the Zamiaceae crown group having a higher rate than the background was not significantly worse (ΔAICc = 0.67). On the other hand, a model with the genus Cycas having a lower rate than the background was significantly worse than the best model (ΔAICc = 2).

### Time series analysis
Our time series analysis did not find a correlation between temperature or $CO_2$ and our disparity metrics through time. All correlations coefficients were lower than 0.5 (Supplementary Fig. 7).

### Discussion
Our analysis does not support the hypothesis of stasis and reduction in cycad leaf disparity through time[15]. The size of the leaf morphospace does instead increase up to the present, with major periods of expansion corresponding to the transition between the Mid and Late Triassic, the Early and Mid-Jurassic, and the Oligocene to the Miocene and present. While the expansion in the Triassic corresponds with an early burst of disparity within the crown-group Cycadales, a common pattern in many clades[19], the one in the Jurassic and the later increases are driven by the origin and expansion of the Zamiaceae. Indeed, during the Early Cretaceous we see the appearance of fossil leaves in the stem groups of *Dioon* and *Bowenia*[20,21], as well as forms with less clear affinities[22]. This strengthens the hypothesis that Zamiaceae underwent an evolutionary radiation during the Jurassic-Cretaceous[16], a pattern suggested by the point estimates of the evolutionary rates in the Early Cretaceous. This radiation would be quasi-contemporaneous to those of other plant groups such as Gnetales[23,24], Podocarpaceae[25], and angiosperms[26,27], suggesting the possibility of a global turnover event across seed plants. Even though our analyses do not seem to indicate that temperature or $CO_2$ were directly driving cycad disparity, it cannot exclude the impact of more complex factors such as aridity. Moreover, some important traits such as the size of the leaves, vein density, or stomatal size and density were not included in our dataset, opening the possibility of a more thorough analysis of the physiological variation of cycads through time. Further investigation on cycads and on the other groups seemingly radiating during the same period should help to test the generalities of this phenomenon and help to disentangle its causes.

Contrary to expectations of a demise of cycads caused by the competition of more efficient and fast-growing flowering plants[15,28], leaf disparity does not seem to decrease in response to the rise and expansion of the angiosperms during the Cretaceous[29]. Recent work has suggested that some Cycadales, namely Zamiaceae and some crown group Cycadaceae, avoided competition with angiosperms by evolving nitrogen fixation, while the non-fixing cycads declined[30]. Our data could seem to partially support

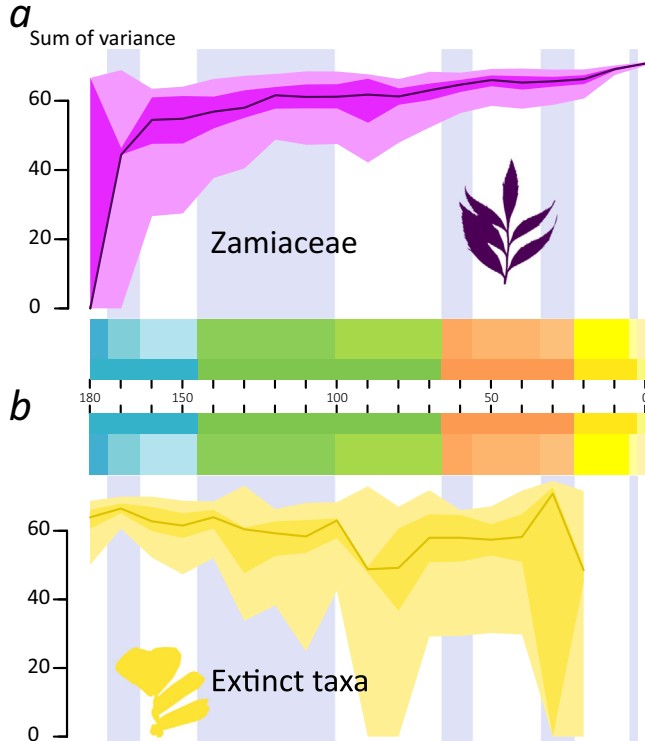

**Fig. 4 | Disparity-Through-Time (DTT) plots of different cycad groups from the Middle Jurassic to the Present. a** DTT plot of the sum of the variance of leaves of the Zamiaceae using the morphospace from the Pre-Ordination Ancestral Reconstruction analysis. The graph shows that in the Zamiaceae there is a fast increase during the early phase of the evolution of the group, followed by a slower increase up to a peak in the Early Cretaceous. **b** DTT plot of the sum of the variance of leaves of extinct cycads, i.e. excluding the crown groups of Zamiaceae and Cycadaceae. In the extinct taxa, it shows a gradual decrease with a low valley in the Late Cretaceous, with the lowest disparity reached before their extinction in the Neogene. The black line indicates the bootstrapped median, light grey indicates the 25–75% percentiles, dark grey indicates the 2.5–97.5% percentiles.

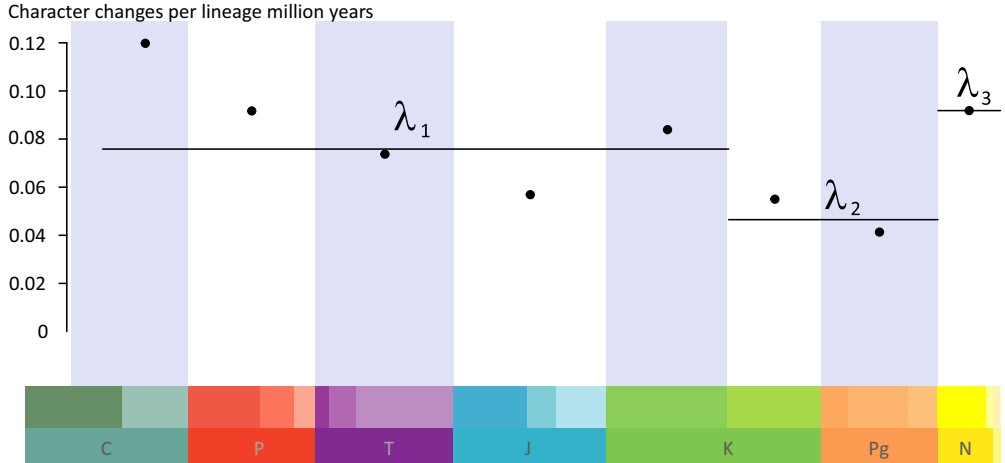

**Fig. 5 | Rates through time plot for all the characters across the cycad phylogeny, showing high rates between the Carboniferous and the Early Cretaceous (λ1), lower rates during the Late Cretaceous and the Paleogene (λ2), and the highest** rates during the Neogene (λ3). Points represent estimates for each time bin, while lines represent estimates from the best model including three rates.

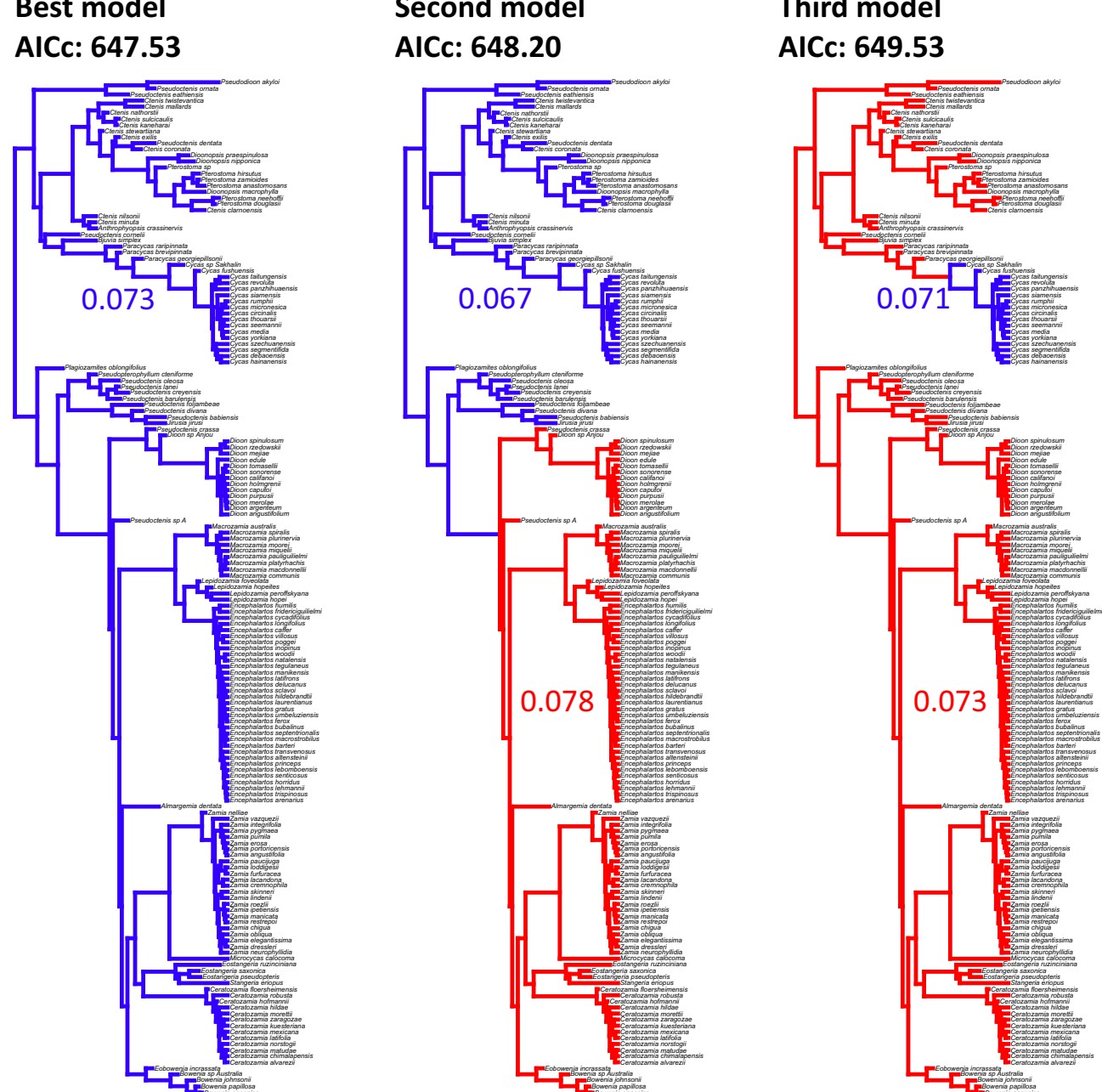

**Fig. 6 | Model for rates of morphological evolution across the tree, testing whether lineages inferred to have acquired nitrogen fixation have different rates than the other cycads.** Though the best model has a single rate across the tree, the second model shows an increased rate in crown group Zamiaceae. A model with Cycadaceae having lower rates is significantly worse than the best model (ΔAICc =2).

this since Zamiaceae show both an increase starting from the Early Cretaceous onwards while the morphospace of other cycads slowly declined (Fig. 4). Moreover, there is an indication (albeit weak) that Zamiaceae have higher rates of morphological evolution compared to the rest of the Cycadales. On the other hand, competition can generate complex dynamics[31], and thus the expectation of a simple decrease in disparity indicating dismissal by competition might be naive. Further investigation on the ecology and ecophysiology of the cycads from the *Ctenis* clade and other extinct groups might help to truly understand the causes of their decline and extinction[32].

The Neogene burst in cycad disparity and evolutionary rates corresponds broadly with the origin of extant species diversity within the extant 10 genera[12,13,17], suggesting that the increase in species diversity correlated

with an increased variation in leaf morphology. This is in agreement with the observation of the variability of morphology and anatomy in some of the extant genera including *Cycas*[33], *Zamia*[34], *Dioon*[35,36], and *Ceratozamia*[11]. The generation of this level of variation in a relatively short time span seems to be at odds with the widespread assumptions on the biology of cycads: these plants have exceedingly long lifespans, small population sizes, and rather slow rates of molecular evolution[28,37]. The availability of genomic data for the cycads[38] and the comparison between genomes of closely related species might help to find the causes of this apparent conundrum, making cycads a potentially fundamental system for understanding the genetic basis for morphological differentiation and adaptive evolution.

Interestingly, no reduction of disparity can be observed in response to major extinction events such as the Permian-Triassic and the Cretaceous-

**Table 1 | Number of data points (tips) per time bin in the Only Taxa analysis**

| Time bin | n |
| --- | --- |
| 252-201.4 | 6 |
| 201.4-145 | 18 |
| 145-66 | 19 |
| 66-23 | 20 |
| 23-0 | 116 |

Palaeogene extinctions. This agrees with other lines of evidence showing a less severe effect of mass extinction on plant groups[39–41]. However, the patterns associated with random or trait-selective extinction events are rather varied and context-dependent[42], and a reduction in disparity might not be observed even after a mass extinction event. Moreover, our sampling strategy may be too coarse to detect the actual response of cycads to these mass extinctions. Further analyses are needed to more confidently test this.

It has to be kept in mind that trends in leaf disparity might not fully capture changes in disparity at the whole-plant level. It has been argued that whole-plant morphology is necessary to investigate the trajectories of disparity through time[43]. However, the fossil record of plants is by its nature fragmentary, and the process of reconstructing whole plants is extremely complex[44]. In the cycads, the issue is complicated by their dioecious nature, with the only whole-plant reconstruction known in some detail only bearing male strobili[45–47]. On the other hand, leaf fossils are relatively well preserved, and thus allow us to sample morphology over time in a more continuous way. Moreover, leaf morphology has important consequences for plant ecology and adaptation[48,49], and thus represents a worthy line of investigation by itself. Based on these considerations, as well as more recent studies approaching single organ disparity[50,51], we think that focusing on single organs can still bring fundamental insights in the macroevolutionary dynamics of plants[50–52].

In conclusion, phylogeny, biogeography, and disparity all agree in showing that the "living fossil" metaphor is inappropriate for the cycads. While metaphors can be powerful tools that drive entire research programmes forward (i.e. the 'adaptive radiation' metaphor[53–55]), they can become deleterious by highlighting some aspects of a group's biology while hiding others that might be as, if not more, important[5]. This seems to be the case with the 'living fossil' metaphor and the cycads: this metaphor has highlighted the case of apparent 'stasis' while hiding the dynamic history of diversification and disparification of this group through time[12,13,17]. We suggest that stasis in cycads should be considered at the level of the single traits[3,56], instead of assuming it based on vague whole-plant morphological similarity, while the dynamic history of this group of "slow-growing", dioecious plants should be the focus of more research at the organismic and molecular level. This new pluralistic view will bring better insights into this charismatic and endangered lineage.

## Material and Methods

### Analysis environment
All analyses were conducted in R ver. 4.2.1.

### Morphological matrix and tree
To analyse cycad leaf disparity through time, we used the matrix and trees from ref. [17]. This tree was generated using a combination of molecular data (15 nuclear loci, one mitochondrial locus, and two plastidial loci) from 321 extant species and morphological data (31 characters) from the extant species and 60 leaf fossil taxa spanning from the Permian to the Miocene. These data were analysed in a Bayesian dated framework using the Fossilized Birth-Death prior[57,58].

The matrix of 31 morphological traits, scored from both direct observation and a review of the large literature on the morphology and anatomy of cycad leaves[20–22,59–91] was expanded to include 14 new characters,

for a total of 45 (Supplementary Methods S1). To avoid issues with high levels of missing data, we reduced the taxon sampling to include only extant taxa that were scored for cuticular characters in the original matrix, leading to 109 extant taxa (from the original 321[6]) and 60 fossil taxa. The extant taxa included all extant genera, with 2 out of 2 species of *Bowenia* Hook. ex Hook.f., 11 out of 40 species of *Ceratozamia* Brongn., 15 out of 119 species of *Cycas*, 13 out of 18 species of *Dioon* Lindl., 33 out of 65 species of *Encephalartos* Lehm., 2 out of 2 species of *Lepidozamia* Regel, 8 out of 41 species of *Macrozamia* Miq., and 23 out of 86 species of *Zamia* L.

The tree was trimmed to only include the 169 species from the morphological matrix using the function drop.tip from the R package ape[92]. A random sample of 100 trees from the posterior distribution from ref. [17] were also used to test the sensitivity of our results to topological uncertainty. All zero-length branches were transformed to 0.001-length branches.

### Morphospace
The function ordinate_cladistic_matrix from the package Claddis[93,94] was used to generate two morphospaces: one was built using only data from the scored taxa, using the prior estimates from[17] as first and last occurrence of the fossil tips, with extant species set to 0-0 (Only Taxa, OT); the other used the tree to conduct ancestral state reconstruction, to increase sample size for the past time bins, thus combating sampling bias (Pre-Ordination Ancestral State Reconstruction, POASR). For both datasets, the distance calculations used the Maximum Observable Rescaled Distance (MORD)[94]. This metric is based on the Gower Coefficient (GC)[95], a measure of distance rescaled by the number of characters that can be scored in both taxa compared. The MORD further rescales the GC to the maximum possible distance, leading to a value between 0 and 1.

The MORD has been shown to perform better than the GC and the General Euclidean Distance (GED) for analyses of disparity. The calculation of the distances was followed by a Principal Coordinate Analysis (PCoA), both operations included in the ordinate_cladistic_matrix function. To test robustness to topological uncertainty, we analysed morphospaces across 100 trees taken from the posterior of ref. [17]. For this analysis, a combination of the function multi.tree and char.diff from the package dispRity[96] and the function pcoa from the package ape[92] was used to replicate the same process followed by the function ordinate_cladistic matrix from the package Claddis[93,94] while optimizing the speed of the analysis.

### Disparity through time
For the Disparity Through Time (DTT) analysis (i.e., the amount of morphological variation across time bins), the OT morphospace was split into time bins equivalent to Periods, given the paucity of data, starting from the Triassic and ending in a combined Miocene-Pliocene-Quaternary bin (from 23 to 0 Ma) (Table 1). This was done using the function chrono.subset from the package dispRity[96]. For the analysis including reconstructed data, the morphospace was split into 10 Ma time bins from 300 Ma to 0 (Table 2). The binning was conducted using the continuous method[96], and assuming both a gradual split model and a punctuated model. Matrices were then bootstrapped for 500 replicates using the function boot_matrix. We also tested the impact of sampling by rarefying the data to 6 and 3 samples.

Following the recommendations presented in ref. [96], we tested the appropriateness of the metrics representing the expansion of cycad leaf morphospace using the web-based shiny app "moms". After this analysis, we selected the sum of variance as an estimate of the size of the morphospace and the mean distance from the centre of the morphospace (the 0,0 point) as a measure of position. These metrics were then calculated using the function dispRity.

### Clade-specific disparity through time
To test the contribution of the different clades to the disparity dynamics of the cycads, we generated DTT curves for the Zamiaceae and for the taxa that are not strongly supported as close relatives of the extant clades. These were defined as taxa or clades that did not result in a clade with Zamiaceae or Cycadaceae with pp=1 in the consensus tree from ref. [17]. DTT for the

**Table 2 | Number of data points (tips plus nodes) per time bin in the Pre-Ordination Ancestral State Reconstruction analysis according to the type of model used to integrate node data (gradual vs punctuated) in the different sampling strategies**

| Time bin | n total (gradual) | n total (puncutated) | n Zamiaceae (gradual) | n Zamiaceae (puncutated) | n Extinct cycads (gradual) | n Extinct cycads (puncutated) |
|---|---|---|---|---|---|---|
| 300 | 3 | 2 | NA | NA | 3 | 2 |
| 290 | 5 | 3 | NA | NA | 5 | 3 |
| 280 | 6 | 4 | NA | NA | 6 | 4 |
| 270 | 6 | 5 | NA | NA | 6 | 5 |
| 260 | 7 | 6 | NA | NA | 6 | 5 |
| 250 | 7 | 6 | NA | NA | 5 | 5 |
| 240 | 9 | 6 | NA | NA | 7 | 5 |
| 230 | 8 | 8 | NA | NA | 6 | 6 |
| 220 | 9 | 8 | NA | NA | 7 | 6 |
| 210 | 12 | 11 | NA | NA | 10 | 9 |
| 200 | 8 | 7 | NA | NA | 6 | 5 |
| 190 | 10 | 9 | NA | NA | 8 | 7 |
| 180 | 13 | 11 | 2 | 1 | 9 | 7 |
| 170 | 19 | 17 | 3 | 3 | 13 | 12 |
| 160 | 14 | 13 | 5 | 4 | 8 | 8 |
| 150 | 13 | 11 | 5 | 4 | 7 | 6 |
| 140 | 17 | 15 | 6 | 5 | 9 | 9 |
| 130 | 12 | 11 | 7 | 6 | 4 | 4 |
| 120 | 17 | 12 | 9 | 7 | 7 | 4 |
| 110 | 14 | 12 | 7 | 6 | 6 | 5 |
| 100 | 14 | 14 | 7 | 6 | 6 | 6 |
| 90 | 12 | 10 | 6 | 6 | 3 | 3 |
| 80 | 16 | 11 | 10 | 7 | 4 | 3 |
| 70 | 17 | 14 | 10 | 8 | 5 | 4 |
| 60 | 18 | 16 | 12 | 10 | 5 | 5 |
| 50 | 20 | 18 | 12 | 12 | 6 | 5 |
| 40 | 17 | 17 | 11 | 11 | 5 | 5 |
| 30 | 17 | 14 | 15 | 11 | 2 | 2 |
| 20 | 22 | 19 | 16 | 12 | 3 | 3 |
| 10 | 53 | 42 | 35 | 32 | NA | NA |
| 0 | 109 | 109 | 94 | 90 | NA | NA |

Cycadaceae could not be reconstructed due to the lack of data caused by the sparse fossil record of the Cycadaceae, including only 5 species over the whole timespan of the crown group leading to only 1 or 2 points per time bin.

The morphospace from the pre-ordination ancestral reconstruction analysis and the tree were trimmed to only include members of either Zamiaceae or completely extinct clades. For the Zamiaceae analysis, we used time bins of 10 Ma from 180 to 0, while for the extinct clade analysis the time bins extended to the age of the tree root.

### Rate analysis

To test the variation in evolutionary rates through geological time, we used the time bin method test_rates in the library Claddis[92]. Time bins were set to be equal to Periods (Carboniferous 330.36-299, Permian 299-252, Triassic 252-201, Jurassic 201-145, Paleogene 66-23) except for the Cretaceous, where we considered the Series level (Early Cretaceous 145-100, Late Cretaceous 100-66), and the combined Neogene-Pliocene-Holocene (23-0). All combinations of rate models between the 8 time bins were tested, for a total of 128 comparisons. The best model was selected using AICc to avoid overparameterization.

We further tested whether the acquisition of nitrogen fixation had an effect on the rates of character evolution in cycads. We used the clade

method in the test_rates function to test whether the two clades that are inferred by[30] to have acquired nitrogen fixation (namely the crown group Zamiaceae and the clade including the Late Cretaceous *Cycas* from Sakhalin (*Cycas* sp. Sakhalin) and extant *Cycas*) had different rates of morphological evolution compared with a model with a single rate across the tree. AICc was used to select the best model.

### Time series analysis

To test the correlation between the disparity of cycad leaves and macroclimatic factors, we downloaded $CO_2$ and temperature data through geological time from ref. [97]. We selected this dataset since it spans the same time span of our analyses, and includes temperature and $CO_2$ estimates in 10 Ma intervals, allowing a direct comparison. To deal with spurious correlation due to strong autocorrelation in time series data, the $CO_2$ and temperature data were analyzed using an Autoregressive Integrated Moving Average (ARIMA) model, implemented in the function auto.arima from the package forecast[98]. The same ARIMA model was then used to analyze the bootstrapped median values for the sum of variance and the centroid distance disparity metrics obtained from the gradual and punctuated models of the POASR analysis using the function Arima from the package forecast. The residuals of the

ARIMA model for temperature or $CO_2$ were then correlated with the residuals of the same ARIMA model applied to the different dependent variables using cross-correlation as implemented in the function ccf from the package stats.

## Reporting summary

Further information on experimental design is available in the Nature Portfolio Reporting Summary linked to this Article.

## Data availability

Data used in this manuscript are available on Figshare: https://figshare.com/s/c0d9432df52860775c77, https://doi.org/10.6084/m9.figshare.23736381.

## Code availability

Scripts used in this manuscript are available on Figshare: https://figshare.com/s/c0d9432df52860775c77, https://doi.org/10.6084/m9.figshare.23736381.

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

## Acknowledgements

This research was funded in whole or in part by the Austrian Science Fund (FWF), grant doi: 10.55776/M3168. The computational results presented have been achieved in part using the Vienna Scientific Cluster (VSC). We are deeply grateful to Thomas Guillerme for implementing functions in dispRity allowing us to analyse data across multiple trees, and further suggestions on the analyses. We also thank Jamie B. Thompson and an anonymous reviewers for comments that greatly improved the manuscript. We also thank Michael Calonje for allowing us to use many of the cycad leaf images included in Fig. 1.

## Author contributions

M.C. and L.J.S. conceived the idea. MC collected the data, analysed the data, generated the graphs and drafted the figures. M.C. and L.S. wrote the manuscript.

## Competing interests

The authors declare no competing interest.
