## [Peer Review File · Communications Biology]

Reviewers' comments:

Reviewer #1 (Remarks to the Author):

The authors provide a brief but fairly compelling account of analyses reconstructing the disparity of Cycads through time, to test whether Cycads are so-called “living fossils”. While this is certainly an interesting read, there are some major issues which must be addressed. These include changes in style, figures and writing, as well as addressing the impact of missing data - this is exemplified by the lack of DTT curve for Cycadaceae.

Overall, I enjoyed this paper, but I have needed to recommend changes to address several major and minor issues. None of my suggested actions are particularly time-consuming though, but I believe they would elevate this study.

The article is quite short, which is not necessarily a bad thing. But in some places, it reads very rushed, like an afterthought based on the authors previously generated data, instead of the high-quality investigation that it could be. It needs further elaboration throughout to elevate it to read more like an independent paper, as well as further details especially in the methods.

In the introduction, I would like to see more detail about Cycads for non-specialists. As an evolutionary botanist, I appreciate the unique position of cycads in evolution and ecology, but many readers may not. I would advise discussing more about their history, distribution and position in ecosystems. Crucially, I would like to see comparisons of diversity/disparity/evolutionary history with the other plant lineages (e.g. Ginkgo, ferns, and especially angiosperms), as well as comparing with other examples of living fossils. Even just a hand-waving comparison would add lots of colour.

I do personally believe the Cycads are the ideal lineage to test the narrative of “living fossils” through analysis of disparity, but this should be explained explicitly, and sold to the general reader. Currently it feels a bit like a study just capitalising on data generated for the previous New Phytologist study by the authors, but it does not have to be - with elaboration and more detail.

The methods are too short and lack detail throughout, and I could not work out some critical things. A major example of this is that the authors mention pruning the data to keep 60 extinct species in line 86, but then say “the tip-only morphospace” in line 103 for the disparity through time. They do appear to have analysed extinct data (lines 93-95), but was this just to reconstruct ancestral states based on tip-variation for later analysis? Or are the extinct data explicitly modelled alongside tip-data? I cannot work it out. This is not clear and needs to be explained in more detail.

Similarly, *Communications Biology* is a general biology journal without specialisation in paleontology, and readers may be unfamiliar with terms such as “MORD distance”, “GED distance”, and even “disparity through time”. Acronyms need defining in full, and explanation in easier terms as to what they do, and what they can reveal, e.g. “maximum observable rescaled distance (MORD), which does xyz to tell us xyz”.

I would like to see details on how the tree was reconstructed previously. Some people may not visit the New Phyt paper in which the FBD tree was reconstructed previously by author one. What molecular and morphological data were sampled? How incomplete was the taxonomic sampling? What methods were used, and time-calibration points? The FBD tree must have taken a remarkable effort to produce, and it deserves describing in this article, to show readers this is a high-quality tree made with state-of-the-art

methods. Paleo-literate readers may possibly think fossil taxa were placed randomly/subjectively via less sophisticated methods, without this detail.

The authors pruned taxa without adequate morphological data for their analyses - I would like to see a description of how uneven the pruned tree was. In the study which originally presented the tree, a genus-level breakdown was provided. This would be very valuable for this study, along with a statement about whether any important species/lineages were not sampled. This would allow the authors to state something like "although sampling was incomplete, we captured the bulk of morphological diversity". I am also highly concerned that there was not enough data for Cycadaceae to model disparity through time, which suggests that a great deal of important species were removed.

A minor concern lies in the use of multi2di to randomly bifurcate the tree. Generally, I do not believe this has much impact on reconstructed dynamics. But I have found personally this can change outcomes, and some authors test a random sample of randomly bifurcated trees in their analyses (e.g. <https://doi.org/10.1111/j.2007.0030-1299.16142.x>). The authors could describe whether the polytomies were or were not crucial/deeper splits, and if they were, may consider testing the robustness of analyses. Similarly, it may be valuable to replicate analyses across a sample of trees from the posterior distribution. But this may not be necessary if authors describe how the FBD tree was well-resolved with well-supported branches and demonstrate that the multi2di would not change results.

Similarly, the current figures presented are fine and do illustrate the results. However, this study would certainly benefit from further figures. E.g., I think figure S2 is an important figure and should be within the main manuscript. Also, the figure legends require more detail explaining explicitly what they mean, rather than just what they show. For instance, instead of saying something akin to "rate variation through time", say "rates increase/decrease/are static"). This is especially important given the living fossil metaphor which may attract readers from outside paleontology.

Crucially, the evidence for this study hinges on leaf variation, yet there is no image of a leaf (except two silhouettes in Figure 2). I believe this necessary for a study into whether cycads are living fossils, not only to add some colour, but to show readers the variation (or lack thereof). Similarly, a figure of a phylogenetic tree would be nice to see, possibly with images of cycads on it (similar to <https://royalsocietypublishing.org/doi/full/10.1098/rsos.140434>).

The authors talk about "overlooked climatic or geological factors" which might have contributed to increase diversification and disparification rates (lines 221-222). I do appreciate the value of a proof-of-concept study in evolution. But analysing this would be very quick and would add so much to this study. I.e., not just saying "there was variation", saying "variation was caused by global temperature/CO2/geographic distribution". They could analyse the relationship between paleoclimatic reconstructions, or whether disparity varies spatially.

Reviewer #2 (Remarks to the Author):

This is a quite interesting well written manuscript. There are some issue that need attention.

1. Why only list characters 32-45 in the supplemental document. Add 1-31.
2. I find many fossil leaves are missing in the matrix. For example, add Austrozamia which has epidermal features share only with Lepiozamia in extants and I think whole preserved leaflets on a rachis.
3. I find it odd that nothing on earlier leaf anatomy and morphology are cited such as Greguss and many other in the bibliography of Norstog and Nicholls. The impression given that all the data originated with the authors. One way to do this might be to use something like [14 and literature cited therein]
4. I find it odd that other paleobotanical works such as Joan Watsons is not cited nor are the taxa included.
5. There is something wrong in the current formatting of the Literature citations. They are quite difficult to parse.

The only thing original here are the analyses and that is very worthwhile. I mainly object to ignoring literature and data sources.

Referee expertise:

Referee #1: Plant macroevolution and evolutionary rates

Referee #2: Cycad morphology, phylogeny and evolution

Reviewers' comments:

Reviewer #1 (Remarks to the Author):

The authors provide a brief but fairly compelling account of analyses reconstructing the disparity of Cycads through time, to test whether Cycads are so-called "living fossils". While this is certainly an interesting read, there are some major issues which must be addressed. These include changes in style, figures and writing, as well addressing as the impact of missing data - this is exemplified by the lack of DTT curve for Cycadaceae. Overall, I enjoyed this paper, but I have needed to recommend changes to address several major and minor issues. None of my suggested actions are particularly time-consuming though, but I believe they would elevate this study.

The article is quite short, which is not necessarily a bad thing. But in some places, it reads very rushed, like an afterthought based on the authors previously generated data, instead of the high-quality investigation that it could be. It needs further elaboration throughout to elevate it to read more like an independent paper, as well as further details especially in the methods.

In the introduction, I would like to see more detail about Cycads for non-specialists. As an evolutionary botanist, I appreciate the unique position of cycads in evolution and ecology, but many readers may not. I would advise discussing more about their history, distribution and position in ecosystems. Crucially, I would like to see comparisons of diversity/disparity/evolutionary history with the other plant lineages (e.g. Ginkgo, ferns, and especially angiosperms), as well as comparing with other examples of living fossils. Even just a hand-waving comparison would add lots of colour.

I do personally believe the Cycads are the ideal lineage to test the narrative of "living fossils" through analysis of disparity, but this should be explained explicitly, and sold to the general reader. Currently it feels a bit like a study just capitalising on data generated for the previous New Phytologist study by the authors, but it does not have to be - with elaboration and more detail.

Reply: We have expanded the introduction to include more details about the cycads and their diversity and trajectories compared to other vascular plants.

"Cycads are a charismatic group both for scientists and the general public. They are a group of gymnosperm plants characterised by a palm-like habit and large compound leaves, as well as having plants of separate sexes (dioecy). The order Cycadales includes around 375 species in two families [6], the Cycadaceae (including the genus Cycas L. and ~ 119 species) and the Zamiaceae (including the other 9 genera and the majority of the species diversity). Nowadays, they are

distributed in tropical and subtropical climates, with major centres of diversity in Mexico and Central America, South Africa, and Australia [7]. Among the extant plants, the diversity and disparity of cycads is dwarfed by other groups such as the angiosperms and the polypodiaceous ferns, but it is similar to that of other groups of comparable age, such as Araucariales (around 200 species) or Gleicheniales (also around 200 species). Even so, cycads are often grouped with Ginkgoales (including the only extant species Ginkgo biloba) as examples of plant living fossils. “

The methods are too short and lack detail throughout, and I could not work out some critical things. A major example of this is that the authors mention pruning the data to keep 60 extinct species in line 86, but then say “the tip-only morphospace” in line 103 for the disparity through time. They do appear to have analysed extinct data (lines 93-95), but was this just to reconstruct ancestral states based on tip-variation for later analysis? Or are the extinct data explicitly modelled alongside tip-data? I cannot work it out. This is not clear and needs to be explained in more detail.

Reply: We see the confusion engendered by our original phrasing. The ‘tip-only’ analysis uses all the data from extant and fossil taxa (since they are all treated as tips in the FDB tree), but does not integrate with reconstructed character states across the phylogeny. The Pre-Ordination Ancestral State Reconstruction analysis used both the information from the taxa and reconstructed character states. We renamed the tip-only analysis to ‘Only Taxa (OT)’ to avoid engendering confusion. We have added further details throughout this section to make it easier to follow.

Similarly, Communications Biology is a general biology journal without specialisation in paleontology, and readers may be unfamiliar with terms such as “MORD distance”, “GED distance”, and even “disparity through time”. Acronyms need defining in full, and explanation in easier terms as to what they do, and what they can reveal, e.g. “maximum observable rescaled distance (MORD), which does xyz to tell us xyz”.

Reply: That is an excellent point. We have now expanded the explanation of these metrics, thus hopefully improving clarity.

“For the Disparity Through Time (DTT) analysis (i.e., the amount of morphological variation across time bins)”

“For both datasets, the distance calculations used the Maximum Observable Rescaled Distance (MORD) [53]. This metric is based on the Gower Coefficient (GC), a measure of distance rescaled by the number of characters that can be scored in both taxa compared. The MORD further rescales the GC to the maximum possible distance, leading to a value between 0 and 1.

The MORD has been shown to perform better than the GC and the General Euclidean Distance (GED) for analyses of disparity. The calculation of the distances was followed by a Principal Coordinate Analysis (PCoA), both operations included in the ordinate_cladistic_matrix function. “

I would like to see details on how the tree was reconstructed previously. Some people may not visit the New Phyt paper in which the FBD tree was reconstructed previously by author one. What molecular and morphological data were sampled? How incomplete was the taxonomic sampling? What methods were used, and time-calibration points? The FBD tree

must have taken a remarkable effort to produce, and it deserves describing in this article, to show readers this is a high-quality tree made with state-of-the-art methods. Paleo-literate readers may possibly think fossil taxa were placed randomly/subjectively via less sophisticated methods, without this detail.

Reply: We have added a more detailed description of the methods used to generate the tree in the method section.

“This tree was generated using a combination of molecular data (15 nuclear loci, one mitochondrial locus, and two plastidial loci) from 321 extant species and morphological data (31 characters) from the extant species and 60 leaf fossil taxa spanning from the Permian to the Miocene. These data were analysed in a Bayesian dated framework using the Fossilized Birth-Death prior [16,17].”

The authors pruned taxa without adequate morphological data for their analyses - I would like to see a description of how uneven the pruned tree was. In the study which originally presented the tree, a genus-level breakdown was provided. This would be very valuable for this study, along with a statement about whether any important species/lineages were not sampled. This would allow the authors to state something like “although sampling was incomplete, we captured the bulk of morphological diversity”.

Reply: we have now added the number of species left after the pruning

“The extant taxa included all extant genera, with 2 out of 2 species of *Bowenia*, 11 out of 40 species of *Ceratozamia*, 15 out of 119 species of *Cycas*, 13 out of 18 species of *Dioon*, 33 out of 65 species of *Encephalartos*, 2 out of 2 species of *Lepidozamia*, 8 out of 41 species of *Macrozamia*, and 23 out of 86 species of *Zamia*.”

I am also highly concerned that there was not enough data for Cycadaceae to model disparity through time, which suggests that a great deal of important species were removed.

Reply: This is not driven by the sampling of extant *Cycas*, but rather the paucity of the fossil record of the Cycadaceae. In our previous phylogenetic analysis, we had included all of the foliage fossils assigned to Cycadaceae that have preserved cuticular anatomy, and found only 5 fossil taxa that are strongly assigned to the stem of the Cycadaceae for the whole timeframe spanning from the Palaeozoic to the Neogene. This sparse dataset does not allow us to obtain any statistic for the disparity of the Cycadaceae, given that each time bin only contains between one and two data points. This is now also explained better in the method section.

“DTT for the Cycadaceae could not be reconstructed due to the lack of data caused by the sparse fossil record of the Cycadaceae, including only 5 species over the whole timespan of the crown group leading to only 1 or 2 points per time bin.”

A minor concern lies in the use of multi2di to randomly bifurcate the tree. Generally, I do not believe this has much impact on reconstructed dynamics. But I have found personally this can change outcomes, and some authors test a random sample of randomly bifurcated trees

in their analyses (e.g. <https://doi.org/10.1111/j.2007.0030-1299.16142.x>). The authors could describe whether the polytomies were or were not crucial/deeper splits, and if they were, may consider testing the robustness of analyses. Similarly, it may be valuable to replicate analyses across a sample of trees from the posterior distribution. But this may not be necessary if authors describe how the FBD tree was well-resolved with well-supported branches and demonstrate that the multi2di would not change results.

Reply: Thank you for this comment, we have looked back at our analyses and actually realised that the resolution of polytomies is not needed in our case, since our trimmed phylogenetic tree has no polytomies. This was a leftover from a previous workflow, and it was helpful to repeat the analyses on other phylogenies with different degrees of resolution. We have also replicated the analyses on 100 trees samples from the posterior distribution of Coiro et al. (2023) thanks to a new implementation of the dispRity function newly added by its creator Dr. Thomas Guillerme at our suggestion. We now show the results in Figure S4 and briefly mention those results.

“To test robustness to topological uncertainty, we analysed morphospaces across 100 trees taken from the posterior of [17]. For this analysis, a combination of the function multi.tree and char.diff from the package dispRity [60] and the function pcoa from the package ape [56] was used to replicate the same process followed by the function ordinate_cladistic matrix from the package Claddis [57,58] while optimizing the speed of the analysis.”

“The results from the POASR analyses are also robust to topological uncertainty (Fig.S4).”

Figure S4:

Similarly, the current figures presented are fine and do illustrate the results. However, this

study would certainly benefit from further figures. E.g., I think figure S2 is an important figure and should be within the main manuscript.

Reply: We have taken your advice and Figure S2 is now included in the text as Figure 3.

Also, the figure legends require more detail explaining explicitly what they mean, rather than just what they show. For instance, instead of saying something akin to “rate variation through time”, say “rates increase/decrease/are static”). This is especially important given the living fossil metaphor which may attract readers from outside paleontology.

Reply: We see how our previous captions were too synthetic. We have now expanded them to include a more detailed explanation:

“Figure 1: Trimmed time-calibrated consensus tree from Coiro et al. (2023) used in this study. Leaves of *Bowenia spectabilis*, *Ceratozamia chimalapensis*, *Stangeria eriopus*, *Zamia imperialis*, *Zamia sp.*, *Encephalartos lehmannii*, *Encephalartos inopinus*, *Macrozamia secunda*, *Dioon edule*, and *Cycas thouarsii* as shown as an example of extant cycad leaf diversity. Fossils of *Eobowenia incrassata*, *Almargemia dentata*, *Pseudoctenis oleosa*, *Bjuvia simplex*, and *Ctenis nathorstii* are shown as examples of cycad fossil leaf diversity. Images are not to scale.”

“Figure 2: Disparity-Through-Time (DTT) plots of cycad leaves, showing a more dynamic pattern than expected from stasis and/or post-Jurassic decline. (a) DTT plot of the sum of variance of the ‘OTtip-only’ analysis, i.e. including only the tips of the cycad tree, showing an increase of the morphospace of cycad leaves up to the present. (b) DTT plot of the mean distance from the centre of the ‘OTtip-only’ analysis, showing the placement of the morphospace moving through time up to the present. (c) DTT plot of the sum of variance of the Pre-Ordination Ancestral Reconstruction analysis. (d) DTT plot of the mean distance from the centre of the pre-ordination ancestral reconstruction analysis. Both (c) and (d) show similar but more complex pattern than (a) and (b). “

“Figure 3: Scatter plots for the first eight Principal Coordinate Analysis axes (PC1-PC8) for Only Taxa morphospace (top) and the Pre-Ordination Ancestral State Reconstruction morphospace (bottom). Tips and nodes for *Zamiaceae* are coloured in purple, *Cycadaceae* in green, and extinct cycads in yellow. The figure shows that the first PCoA axis (PC1) separates *Zamiaceae* from the rest of the cycads, and that in general the three groups of cycads occupy different parts of the morphospace. On the bottom graphs, the root node is coloured in red, and the phylogenetic relationships are indicated with grey lines.”

“Figure 4: Disparity-Through-Time (DTT) plots of different cycad groups from the Middle Jurassic to the Present. (a) DTT plot of the sum of variance of leaves of the *Zamiaceae* using the morphospace from the pre-ordination ancestral reconstruction analysis. The graph shows that in the *Zamiaceae* there is a fast increase during the early phase of the evolution of the group, followed by a slower increase up to a peak in the Early Cretaceous. (b) DTT plot of the sum of variance of leaves of extinct cycads, i.e. excluding the crown groups of *Zamiaceae* and *Cycadaceae*. In the extinct

taxa, it shows a gradual decrease with a low valley in the Late Cretaceous, with the lowest disparity reached before their extinction in the Neogene."

"Figure 5: Rates through time plot for all the characters across the cycad phylogeny, showing high rates between the Carboniferous and the Early Cretaceous (λ_1), lower rates during the Late Cretaceous and the Paleogene (λ_2), and the highest rates during the Neogene (λ_3). Points represent estimates for each time bin, while lines represent estimates from the best model including three rates."

"Figure 6: Model for rates of morphological evolution across the tree, testing whether lineages inferred to have acquired nitrogen fixation have different rates than the other cycads. Though the best model has a single rate across the tree, the second model shows an increased rate in crown group Zamiaceae. A model with Cycadaceae having lower rates is significantly worse than the best model ($\Delta AICc = 2$)."

Crucially, the evidence for this study hinges on leaf variation, yet there is no image of a leaf (except two silhouettes in Figure 2). I believe this necessary for a study into whether cycads are living fossils, not only to add some colour, but to show readers the variation (or lack thereof). Similarly, a figure of a phylogenetic tree would be nice to see, possibly with images of cycads on it (similar to <https://royalsocietypublishing.org/doi/full/10.1098/rsos.140434>).

Reply: We take the point and we have added a new figure one that includes both the tree and examples of extant and fossil cycad leaves, now figure 1.

The authors talk about “overlooked climatic or geological factors” which might have contributed to increase diversification and disparification rates (lines 221-222). I do appreciate the value of a proof-of-concept study in evolution. But analysing this would be very quick and would add so much to this study. I.e., not just saying “there was variation”, saying “variation was caused by global temperature/CO₂/geographic distribution”. They could analyse the relationship between paleoclimatic reconstructions, or whether disparity varies spatially.

Reply: We have now added two new sets of analyses to have a more concrete first test of these hypotheses. First, we tested whether the acquisition of nitrogen fixation recently analysed in fossil cycads and reconstructed across the Coiro et al. (2023) tree by Kipp et al. (2023, *Nature Ecology and Evolution*) has an impact on the rates of character evolution. These results are now presented in a new figure 6. Moreover, we tested whether temperature and CO₂ concentration correlated with the disparity of cycad leaves. These results are mentioned and shown in the new figure S5. We have also changed our discussion section to deal with the results from Kipp et al. (2023),

which we almost predicted in our previous version. We hope that these new preliminary results will have an impact on future cycad research.

“We further tested whether the acquisition of nitrogen fixation had an effect on the rates of character evolution in cycads. We used the clade method in the test_rates function to test whether the two clades that are inferred by [62] to have acquired nitrogen fixation (namely the crown group Zamiaceae and the clade including the Late Cretaceous Cycas from Sakhalin (Cycas sp. Sakhalin) and extant Cycas) had different rates of morphological evolution compared with a model with a single rate across the tree. AICc was used to select the best model. to have acquired nitrogen fixation (namely the crown group Zamiaceae and the clade including the Late Cretaceous Cycas from Sakhalin (Cycas sp. Sakhalin) and extant Cycas) had different rates of morphological evolution compared with a model with a single rate across the tree. AICc was used to select the best model.

7. Time Series Analysis

To test the correlation between the disparity of cycad leaves and macroclimatic factors, we downloaded CO2 and temperature data through geological time from [63]. We selected this dataset since it spans the same timespan of our analyses, and includes temperature and CO2 estimates in 10 Ma intervals, allowing a direct comparison. To deal with spurious correlation due to strong autocorrelation in time series data, the CO2 and Temperature data were analyzed using an Autoregressive Integrated Moving Average (ARIMA) model, implemented in the function auto.arima from the package forecast [64]. The same ARIMA model was then used to analyze the bootstrapped median values for the sum of variance and the centroid distance disparity metrics obtained from the gradual and punctuated models of the POASR analysis using the function Arima from the package forecast. The residuals of the ARIMA model for temperature or CO2 were then correlated with the residuals of the same ARIMA model applied to the different dependent variables using cross-correlation as implemented in the function ccf from the package stats.

”

“The analysis of rates in different clades corresponding to the acquisition of nitrogen fixation favored a model with a single rate across the tree (Fig. 6). However, a model with the Zamiaceae crown group having a higher rate than the background was not significantly ($\Delta AICc = 0.67$). On the other hand, a model with the genus Cycas having a lower rate than the background was significantly worse than the best model ($\Delta AICc = 2$).

Time series analysis

Our time series analysis did not find correlation between temperature or CO2 and our disparity metrics through time. All correlations coefficients were lower than 0.5 (Fig. S6).

“

Figure 6:

Best model
AICc: 647.53

Second model
AICc: 648.20

Third model
AICc: 649.53

Figure S6:

Temperature

CO₂

“Our analysis does not support the hypothesis of stasis and reduction in cycad leaf disparity through time [15]. The size of the leaf morphospace does instead increase up to the present, with major periods of expansion corresponding to the transition between the Mid and Late Triassic, the Early and Mid Jurassic, and the Oligocene to the Miocene and present. While the expansion in the Triassic corresponds with an early burst of disparity within the crown-group Cycadales, a common pattern in many clades [65], the one in the Jurassic and the later increases are driven by the origin and expansion of the Zamiaceae. Indeed, during the Early Cretaceous we see the appearance of fossil leaves in the stem groups of *Dioon* and *Bowenia* [26,32], as well as forms with less clear affinities [23]. This strengthens the hypothesis that Zamiaceae underwent an evolutionary radiation during the Jurassic-Cretaceous [16], a pattern suggested by the point estimates of the evolutionary rates in the Early Cretaceous. This radiation would be quasi-contemporaneous to those of other plant groups such as Gnetales [66,67], Podocarpaceae [68], and angiosperms [69,70], suggesting the possibility of a global turnover event across seed plants. Even though our analyses do not seem to indicate that temperature or CO₂ were directly driving cycad disparity,

it cannot exclude the impact of more complex factors such as aridity. Moreover, some important traits such as size of the leaves, vein density, or stomatal size and density were not included in our dataset, opening the possibility of a more thorough analysis of the physiological variation of cycads through time. Further investigation on cycads and on the other groups seemingly radiating during the same period should help to test the generalities of this phenomenon and help to disentangle its causes.

Contrary to expectations of a demise of cycads by the competition of more efficient and fast-growing flowering plants [15,71], leaf disparity does not seem to decrease in response to the rise and expansion of the angiosperms during the Cretaceous [72]. Recent work has suggested that some Cycadales, namely Zamiaceae and some crown group Cycadaceae, avoided competition with angiosperms by evolving nitrogen fixation, while the non-fixing cycads declined. Our data could seem to partially support this, since Zamiaceae show both an increase starting from the Early Cretaceous onwards while the morphospace of other cycads slowly decline (Fig. 4). Moreover, there is an indication (albeit weak) that Zamiaceae have higher rates of morphological evolution compared to the rest of the Cycadales. On the other hand, competition can generate complex dynamics [76], and thus the expectation of a simple decrease in disparity indicating dismissal by competition might be naive. Further investigation on the ecology and ecophysiology of the cycads from the Ctenis clade and other extinct groups might help to truly understand the causes of their decline and extinction are retrieved at higher latitudes than extant cycads, and probably had a different ecology [77].”

Kipp, M.A., Stüeken, E.E., Strömberg, C.A., Brightly, W.H., Arbour, V.M., Erdei, B., Hill, R.S., Johnson, K.R., Kvaček, J., McElwain, J.C. and Miller, I.M., 2023. Nitrogen isotopes reveal independent origins of N₂-fixing symbiosis in extant cycad lineages. *Nature Ecology & Evolution*, pp.1-13.

Reviewer #2 (Remarks to the Author):

This is a quite interesting well written manuscript. There are some issue that need attention.

1. Why only list characters 32-45 in the supplemental document. Add 1-31.

Reply: You are right, we have now included all characters in the supplementary document.

2. I find many fossil leaves are missing in the matrix. For example, add *Austrozamia* which

has epidermal features share only with *Lepiozamia* in extants and I think whole preserved leaflets on a rachis.

Reply: the sampling was based on the completeness of the fossil and the availability of stomatal characters. That's why unfortunately *Austrozamia* and many others were not included in the analysis from Coiro et al. 2023.

3. I find it odd that nothing on earlier leaf anatomy and morphology are cited such as Greguss and many other in the bibliography of Norstog and Nicholls. The impression given that all the data originated with the authors. One way to do this might be to use something like [14 and literature cited therein]

Reply: we are sorry for this omission, it was not our intention to give such impression. We have now added a more clear reference to the literature that Coiro et al. 2023 is based on.

4. I find it odd that other paleobotanical works such as Joan Watsons is not cited nor are the taxa included.

Reply: the taxa from Watsons work are indeed included, i.e. *Ctenis twistevantica*, *Ctenis mallards*, *Pseudoctenis foljambeae*. We have now added the citation of Watsons work in the text in the context of the previous comment.

5. There is something wrong in the current formatting of the Literature citations. They are quite difficult to parse.

Reply: unfortunately, we are restricted by the format of Communications Biology. We have now made the references one-and-a-half-spaced, hopefully this will make it easier to parse them in the revised document.

The only thing original here are the analyses and that is very worthwhile. I mainly object to ignoring literature and data sources.

Reply: again, we are sorry for this omission, we agree that we were relying too much on our previous analysis (where the literature is clearly cited) as space was a concern here and I hope we have now corrected this issue.

REVIEWERS' COMMENTS:

Reviewer #1 (Remarks to the Author):

Firstly, I want to say thank you to the authors for a very thorough and prompt response. My concerns have been addressed in a very well-done manner. There are just a few minor corrections suggested at the end of this review.

I appreciate all the responses to my comments. The introduction is now a much better sell for a general-biology audience, the methods are significantly clearer, and acronyms defined. The sources of the data and their limitations described with adequate and honest detail. Thank you for explaining the issue with the fossil record relating to the DTT analyses, the lack of data is unavoidable, and the transparency and clarity has been made clear. Great work replicating the analyses over a sample of 100 of trees too.

Also - figure 1 is stunning now, wow! I appreciate that the work on climate-driven evolution has been greatly improved, it adds so much to the study and elevates the narrative, and will attract a wider-readership.

This manuscript was always compelling, but the previous version lacked detail and read more like an afterthought capitalising on the previous Cycads paper. Now, it is a standalone paper and I thoroughly recommend it for publication in Communications Biology. It asks and answers some very interesting questions in a thorough manner.

Minor corrections:

Do correct line 52, the first line of the second introduction paragraph. There is a rogue comma.

Citation 86 is incorrect as it has been published in Biology Letters, and the title has changed.

One of the figures has the word "extinct" rendered as "ExĈnct".

Reviewer #1 (Remarks to the Author):

Firstly, I want to say thank you to the authors for a very thorough and prompt response. My concerns have been addressed in a very well-done manner. There are just a few minor corrections suggested at the end of this review.

I appreciate all the responses to my comments. The introduction is now a much better sell for a general-biology audience, the methods are significantly clearer, and acronyms defined. The sources of the data and their limitations described with adequate and honest detail. Thank you for explaining the issue with the fossil record relating to the DTT analyses, the lack of data is unavoidable, and the transparency and clarity has been made clear. Great work replicating the analyses over a sample of 100 of trees too.

Also - figure 1 is stunning now, wow! I appreciate that the work on climate-driven evolution has been greatly improved, it adds so much to the study and elevates the narrative, and will attract a wider-readership.

This manuscript was always compelling, but the previous version lacked detail and read more like an afterthought capitalising on the previous Cycads paper. Now, it is a standalone paper and I thoroughly recommend it for publication in Communications Biology. It asks and answers some very interesting questions in a thorough manner.

Reply: Thank you very much for your points, they helped to improve the manuscript significantly.

Minor corrections:

Do correct line 52, the first line of the second introduction paragraph. There is a rogue comma.

Citation 86 is incorrect as it has been published in Biology Letters, and the title has changed. One of the figures has the word "extinct" rendered as "ExCnct".

Reply: These have been dealt with.